# Effects of Eutrophication on Plankton Abundance and Composition in the Gulf of Gabès (Mediterranean Sea, Tunisia)

Neila Annabi-Trabelsi [1,†], Wassim Guermazi [1,†], Vincent Leignel [2], Yousef Al-Enezi [3], Qusaie Karam [3], Mohammad Ali [3], Habib Ayadi [1] and Genuario Belmonte [4,*]

1   Laboratoire Biodiversité Marine et Environnement (LR18ES30), Université de Sfax, Sfax CP 3000, Tunisia; neila.trabelsi@isbs.usf.tn (N.A.-T.); wassim016@yahoo.fr (W.G.); habibayadi62@yahoo.fr (H.A.)
2   Laboratoire BIOSSE, Le Mans Université, 72000 Avenue O Messiaen, France; vincent.leignel@univ-lemans.fr
3   Environment & Life Sciences Research Center, Kuwait Institute for Scientific Research, Safat 13109, Kuwait; yaenezi@kisr.edu.kw (Y.A.-E.); qkaram@kisr.edu.kw (Q.K.); mohammad.awad.ali@gmail.com (M.A.)
4   Laboratory of Zoogeography and Fauna, University of the Salento, 73100 Lecce, Italy
*   Correspondence: genuario.belmonte@unisalento.it
†   These authors contributed equally to this work.

**Highlights:**

- A total of 42 phytoplankton and 24 Copepoda taxa were identified in the coastal environment of Gabès (Tunisia).
- Bacillariophyta were the most abundant group (67.7–89.2% of total phytoplankton abundance).
- Shannon–Weaver diversity index of phytoplankton and Copepoda showed negative correlations with Eutrophication Index.
- Opportunistic and *r* strategist species (*Chaetoceros costatus*, *Euglena acusformis*, and *Thalassiosira* sp.) showed a positive correlation with Eutrophication Index.
- Among Copepoda, *Oithona similis* and *Euterpina acutifrons* were indifferent to eutrophication.

**Abstract:** Phytoplankton and Copepoda were investigated in the industrialized Gabès coast (Mediterranean Sea, Tunisia) to understand consequences of the Eutrophication Grade on the community composition. In the past 40 years, intensive agrochemical activities have developed in Gabès, discharging wastewater along the coast. In parallel, Gabès City has experienced a rapid demographic expansion (population: 131,000) that further increased sewage discharged into the sea. The present study was carried out in the Gulf of Gabès in March 2014. The abiotic analysis of seawater showed high concentrations of nutrients and eutrophication in all the studied fifteen stations. A growing eutrophic gradient was revealed from Zarrat to Gannouche. During this study, 42 phytoplankton taxa and 24 Copepoda taxa were identified. Bacillariophyta were the most abundant group, ranging from 67.7% to 89.2% of total phytoplankton specimens. *Chaetoceros costatus*, *Euglena acusformis*, and *Thalassiosira* sp. showed a positive correlation with Eutrophication Index (profited of nutrient availability). Therefore, the Shannon–Weaver diversity index of phytoplankton and Copepoda showed negative correlations with Eutrophication Index. The relatively high H′ values for phytoplankton suggest that the eutrophicated waters of Gulf of Gabès are not a hostile environment for them. Among Copepoda, *Oithona similis*, and *Euterpina acutifrons* seem to be insensible species to eutrophication.

**Keywords:** Eutrophication Index; phytoplankton; copepoda; diversity

## 1. Introduction

Eutrophication is defined as an increase in the rate of supply of organic matter to an ecosystem [1,2]. Coastal eutrophication caused by nutrient inputs deriving from human activities is one of the greatest threats to the health of coastal estuarine and marine ecosystems worldwide [3]. On a global scale, the increasing rate of coastal urbanization, combined

with climate change, amplifies coastal eutrophication [4]. Eutrophication has been reported to cause an increase in phytoplankton biomass and to induce a selection of tolerant and opportunistic species [5]. Changes in the composition of the primary producers lead to shifts in the zooplankton community [6]. In general, plankton are very sensitive to changes in the environment and they are good indicators of water quality and trophic state [7,8]. For example, owing to their rapid response to environmental conditions and their short life cycles, Copepoda are considered good animal bioindicators of the ecosystem status and health [6,9,10].

Eutrophication is enhanced in confined environments where it is coupled with a reduction in Copepoda species [11], often with the replacement of large-sized species with smaller ones [12–14], or even larger to smaller sized specimens of the same species [15]. The general deviation to small-sized species with increasing confinement (synergic with eutrophication) has been reported to be associated with changes in the food quality [6]. Copepod survival, fitness, sex ratio, and condition are negatively correlated with Cyanobacteria [16] that characterize eutrophic marine ecosystem [17]. Copepod male survival may be more successful than female in eutrophic conditions, and a positive correlation between phosphates and the sex ratio in the copepod was observed by Krupa [18].

The impacts of confinement-eutrophication on biodiversity may also lead to alterations in biological interactions, trophic structure, and primary productivity, with consequences for ecosystem function and services provision [19].

The Gabès area, in Southern Tunisia, is considered one of the most human impacted coastal areas in the Mediterranean Sea [20]. Since the foundation of one of the biggest Tunisian industrial complexes in Gabès City in 1972, the coastal marine ecosystem has been continually polluted by numerous contaminants such as industrial effluents and sewage [21–24]. Phosphogypsum (PG) coming from the Tunisian Chemical Group (TCG), used in the treatment of the phosphates, represents one of the major contaminants [20,25–27]. It is also considered among the plausible major dangers threatening the vulnerable marine ecosystem in this area [21–24,28]. Estimates of phosphogypsum released into the sea range from 1000 to 13,000 tons daily without any treatment [25,29]. Phosphogypsum contains several pollutants such as heavy metals, fluorine, phosphorus, and even radionuclides generating radioactivity [30,31]. As a nutrient, phosphorus has the potentiality to elevate organic matter input to the seabed and consequently to induce eutrophication [32].

The objective of the present study was to find correlations between eutrophication (defined by the use of a Eutrophication Index) and plankton composition (phytoplankton and Copepoda), analyzing diversity, spatial distribution, and abundance of species over an extended grid of stations in the Gulf of Gabès.

## 2. Material and Methods

### 2.1. Field Sampling

The study was carried out during spring 2014, the season when the plankton peak is maximum in the Gulf of Gabès [33,34]. Samples were collected on 15 March 2014 from 15 stations (at water columns varying from 9 to 27 m) grouped in five transects, orthogonal to the coastline along about 30 km. The sampling grid was chosen to cover Gabès coast area as shown in Figure 1.

At each station, samples for physico-chemical analyses, Chl-*a*, and for analysis of phytoplankton were collected with a Van Dorn bottle at 1 m below the sea level. Mesozooplankton were collected using a cylindro-conical net (30 cm mouth, 100 cm length, 100 μm mesh size) equipped with a flowmeter (438 115, HydroBios, Altenholz, Germany). Mesozooplankton hauls were vertical (bottom-surface). The volume of water filtered ranged between 0.6 and 1.9 $m^3$. Samples for nutrient analyses (120 mL) were preserved immediately upon collection ($-20$ °C, in the dark); those for phytoplankton analyses (1 L) were preserved with Lugol (4%) Iodine solution (acid Lugol solution). Mesozooplankton samples were rapidly preserved in 2% buffered formaldehyde solution after collection

and were stained with Rose Bengal to identify internal tissues of different zooplankton species and also to facilitate identification of Copepoda. Water samples (1 L) for Chl-*a* analysis were filtered by vacuum filtration onto GF/F glass fiber filters (Whatman 1825-04, Whatman, Brentford, United Kingdom), which were then immediately stored at −20 °C.

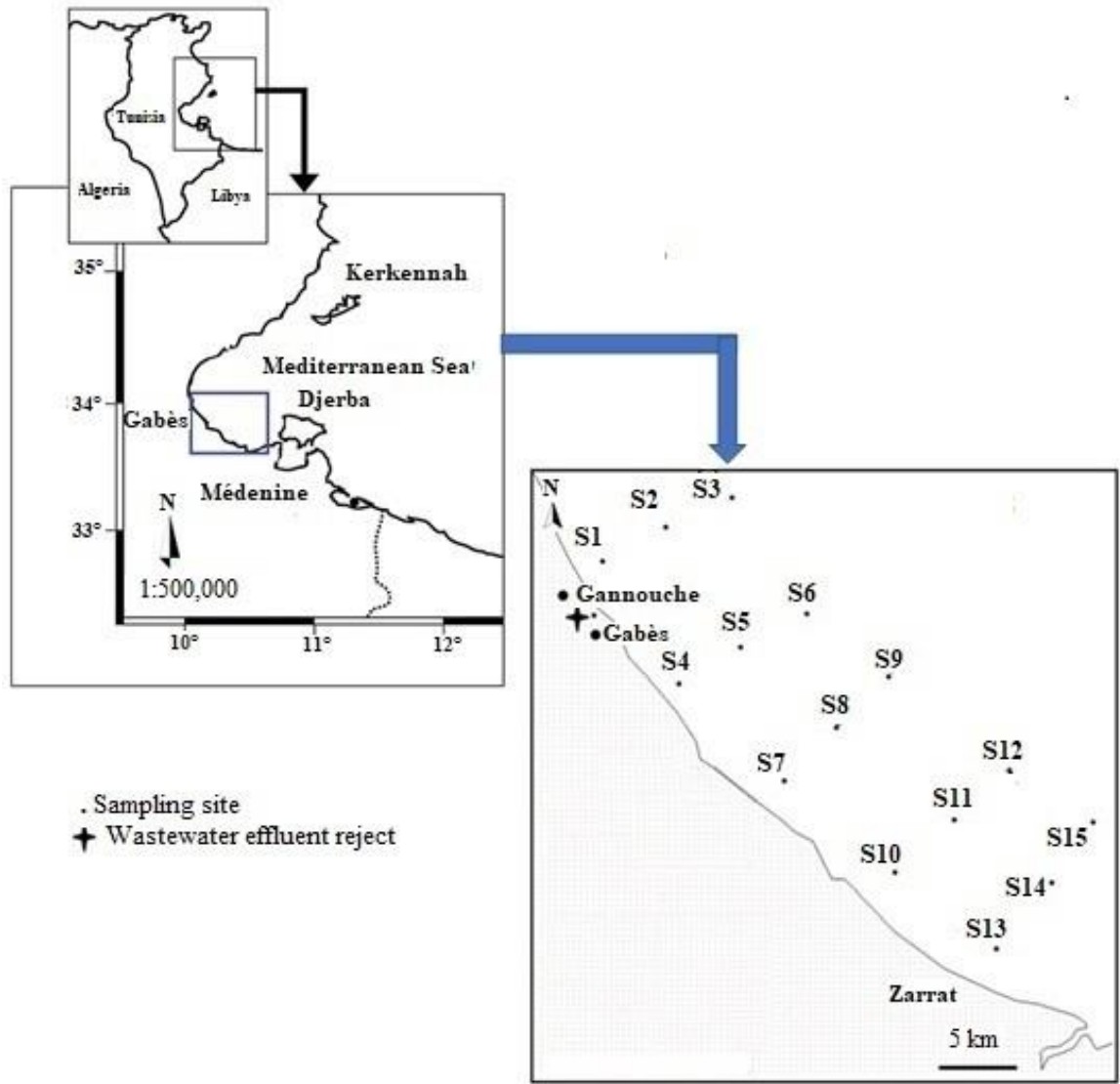

**Figure 1.** Location of sampling stations (S1–S15) along 5 transects perpendicular to the coastline of the Gulf of Gabès.

## 2.2. Physico-Chemical and Chl-a Analysis

Temperature, salinity, and pH were measured immediately after sampling using a multi-parameter kit (Multi 340 i/SET, WTW, Weilheim, Germany). Nutrients (nitrites, nitrates, ammonium, orthophosphates, and silicate) measured were used for the calculation of the Eutrophication Index of Primpas et al. [35]; concentrations were determined with an autoanalyzer (type 3, Bran+Luebbe GmbH, Frankfurt, Germany). Chl-*a* was estimated by spectrometry, after extraction of the pigments in acetone (90%). The concentrations were then estimated using the equations of SCOR-UNESCO (1966).

The Eutrophication Index (E.I.) of Primpas et al. [35] was used in order to assess the eutrophication status of the coastal waters of Gabès. The formula takes into consideration nitrites, nitrates, ammonium, phosphates, and Chl-*a*, resulting in a five-scale scheme according to the WFD (Water Framework Directive) requirements, following different values:

1. (High ecological water quality, E.I. < 0.04),
2. (Good E.I. = 0.04–0.38),
3. (Moderate, E.I. = 0.38–0.85),
4. (Poor, E.I. = 0.85–1.51),
5. (Bad, E.I. > 1.51).

Formula:

$$\text{E.I.} = (0.279 \times \text{phosphates}) + (0.261 \times \text{nitrates}) + (0.296 \times \text{nitrites}) + (0.275 \times \text{ammonium}) + (0.214 \times \text{Chl } a)$$

### 2.3. Phytoplankton and Zooplankton Abundance

Subsamples (50 mL) of phytoplankton were analyzed under an inverted microscope (Leica DM IL, Leica Microsystems, Wetzlar, Germany) using the Utermöhl method after sedimentation for 24 to 48 h (Utermöhl, 1958). Counts were carried out on the entire sedimentation chamber at $400\times$ magnification for phytoplankton. Phytoplankton identification was based on morphological criteria after consulting various keys [36,37].

Zooplankton samples, were identified according to Rose [38] and Boxshall and Halsey [39]. Enumeration was performed under a vertically mounted deep-focus dissecting microscope (Olympus TL 2) at $100\times$ magnification. Copepods were sorted into four demographic classes (nauplii, copepodites, adult males, and adult females) and species were identified only for adults. Copepoda nauplii were considered as a single group; copepodites were counted and grouped according to the order (Calanoida, Cyclopoida, Harpacticoida).

According to Harris et al. [40], diversity indices were used to describe the quality of the community, which depends on the number of species and their relative abundances in each sample. This study adopted Shannon–Weaver diversity index [41], because this index is more sensitive for rare species [42], and $J$, Pielou's evenness index [43].

### 2.4. Statistics

Correlations between variables were calculated when necessary, using Spearman's rank correlation coefficient. This test was chosen because of non-normal distribution of the data. A single factor analysis of variance (one-way ANOVA) completed by Tukey post hoc was conducted to assess significant difference between the five transects sampled with regard to physico-chemical and biological parameters. The significance level was set at 0.05.

In order to determine the group and species patterns, the abundance data of all stations were analyzed using cluster and multidimensional scaling (MDS) techniques, based on the Bray–Curtis similarity index (group average technique). The relationship between environmental variables and patterns of similarity of the phytoplankton and copepods communities was assessed by the MDS plot.

## 3. Results

### 3.1. Seawater Analysis and Trophic State

Water temperature ranged from 16.5 to 19.7 °C (mean $\pm$ s.d. = 18.4 $\pm$ 0.62 °C) (Figure 2), tending to stability over the spatial scale. Salinity ranged from 36 to 40 psu (mean $\pm$ s.d. = 38.80 $\pm$ 1.2), and was high in all sampling stations except S1 (Figure 2). pH varied from 7.68 to 7.83, at stations S1 and S14, respectively. S1, followed by S2, was characterized by the lowest salinity and pH as a result of liquid effluents of TCG and urban wastewaters. While temperature and salinity did not change significantly between the five transects ($p > 0.05$), pH seemed to be significantly affected from T1 to T5 (F = 7.861; $p < 0.01$) (Table 1).

Nitrites, nitrates, ammonium, and phosphates decreased significantly ($p < 0.01$) from Gannouche to Zarrat (Figure 3, Table 1). Nitrates, ammonium, and phosphates were observed at high concentrations in all stations and transects and reflected a eutrophic state (Figure 3). Ammonium was the most dominant nitrogen form, and its concentration ranged from 2.8 to 20.6 μmol $L^{-1}$ (average 8.73 $\pm$ 5.41), recorded at stations S15 and S1, respectively (Figure 3). Ammonium concentrations were significantly higher in T1 and T2, which are in

front of the TCG plant, than those recorded in the southern transects, reaching 17.07 ± 3.60 and 10.63 ± 4.59 μmol L$^{-1}$ (Table 1).

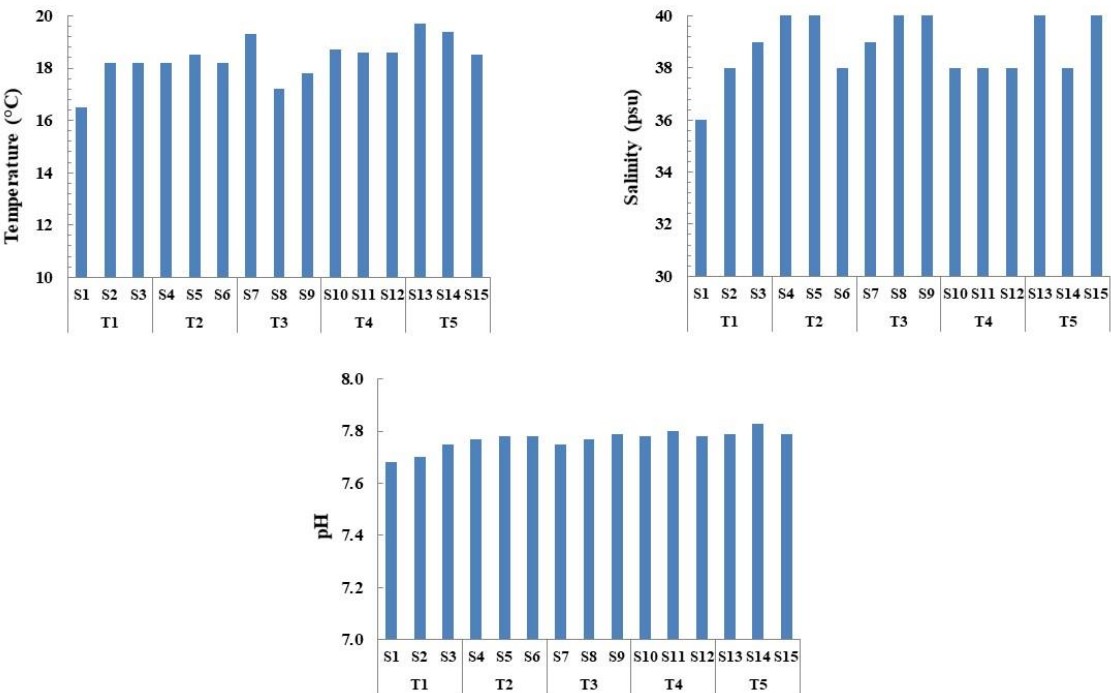

**Figure 2.** Spatial variation of temperature, salinity, and pH at sampled stations (S1–S15), divided according to transects (T1–T5).

**Table 1.** Mean values (Standard Deviation) of physico-chemical and biological parameters of 5 transects sampled in the coastline of the Gulf of Gabès. In the last column are results of one-way ANOVA for the comparison between the five sampled transects followed by Tukey's post hoc test.

| | T1 | T2 | T3 | T4 | T5 | F-Values (d.f) | *p*-Values |
|---|---|---|---|---|---|---|---|
| | | | Physical parameters | | | | |
| Temperature | 17.63 (±0.98) a | 18.30 (±0.17) a | 18.10 (±1.08) a | 18.63 (±0.06) a | 19.20 (±0.62) a | 2.022(14) | 0.167 |
| Salinity | 37.67 (±1.53) a | 39.33 (±1.15) a | 39.67 (±0.58) a | 38.00 (±0.00) a | 39.33 (±1.15) a | 2.281(14) | 0.132 |
| pH | 7.71 (±0.04) a | 7.78 (±0.01) b | 7.77 (±0.02) ab | 7.79 (±0.01) b | 7.80 (±0.02) b | 7.861(14) | 0.004 ** |
| | | | Chemical parameters | | | | |
| Nitrites | 0.67 (±0.06) a | 0.63 (±0.06) ab | 0.53 (±0.06) abc | 0.50 (±0.00) bc | 0.43 (±0.06) c | 10.375(14) | 0.001 ** |
| Nitrates | 5.47 (±0.55) a | 4.90 (±0.30) ab | 4.53 (±0.32) ab | 3.93 (±0.32) bc | 3.43 (±0.29) c | 13.959(14) | 0.000 *** |
| Ammonium | 17.07 (±3.60) a | 10.63 (±4.59) ab | 6.63 (±0.98) b | 4.43 (±0.95) b | 4.90 (±1.84) b | | 0.001 ** |
| Phosphates | 6.17 (±0.61) a | 4.73 (±0.75) b | 3.33 (±0.38) c | 2.67 (±0.21) cd | 1.80 (±0.17) d | 38.876(14) | <0.0001 *** |
| Silicates | 4.63 (±1.89) a | 4.70 (±1.65) a | 5.03 (±1.50) a | 4.23 (±1.14) a | 4.60 (±1.37) a | 0.104(14) | 0.979 |
| E.I. | 7.85 (±0.63) ab | 5.71 (±1.66) ab | 4.32 (±0.74) b | 3.33 (±0.29) b | 3.32 (±0.60) b | 13.333(14) | 0.001 ** |
| | | | Biological parameters | | | | |
| Total-copepods | 40.22 (±9.43) a | 59.17 (±7.82) ab | 95.56 (±15.27) bc | 144.75 (±8.07) c | 265.82 (±32.81) d | 79.984(14) | <0.0001 *** |
| Adults | 11.32 (±6.75) a | 15.26 (±1.16) a | 30.90 (±6.43) b | 38.05 (±2.43) b | 37.63 (±1.25) b | 24.808(14) | <0.0001 *** |
| Copepodids | 4.84 (±1.12) a | 14.31 (±5.87) ab | 28.05 (±4.83) bc | 40.12 (±7.30) c | 57.67 (±6.06) d | 43.923(14) | <0.0001 *** |
| Nauplii | 24.06 (±2.80) a | 29.60 (±3.43) a | 36.61 (±5.60) a | 66.58 (±6.77) a | 170.52 (±37.63) b | 36.866(14) | <0.0001 *** |
| H' copepoda | 2.15 (±0.42) a | 2.55 (±0.05) ab | 2.63 (±0.18) ab | 2.74 (±0.12) ab | 2.91 (±0.16) b | 4.838(14) | 0.020 * |
| Total-Phytoplankton | 117.67 (±41.70) a | 58.63 (±26.80) ab | 27.17 (±6.44) b | 23.60 (±6.15) b | 36.60 (±3.91) b | 8.839(14) | 0.003 ** |
| Bacillariophyta | 48.30 (±17.53) a | 24.50 (±9.73) ab | 14.23 (±2.65) b | 14.37 (±4.46) b | 23.87 (±2.36) ab | 6.677(14) | 0.007 ** |
| Dinophyta | 5.33 (±2.29) a | 2.00 (± 0.61) ab | 1.77 (±0.15) b | 1.57 (±0.55) b | 3.57 (±0.81) ab | 5.824(14) | 0.011 * |
| Euglenophyceae | 11.87 (±7.08) a | 4.23 (±1.91) ab | 1.47 (±0.42) b | 1.13 (±0.25) b | 0.53 (±0.35) b | 6.130(14) | 0.009 ** |
| Cyanobacteria | 3.00 (±0.52) a | 4.20 (±3.82) a | 0.40 (±0.69) a | 0.00 (0.00) a | 0.00 (0.00) a | 3.735(14) | 0.041 * |
| H' Phytoplankton | 2.87 (±0.10) a | 2.92 (±0.19) a | 3.34 (±0.30) ab | 3.52 (±0.23) b | 3.24 (±0.11) ab | 5.816(14) | 0.011 * |

Values in each line that share the same letter are not significantly different (*p* > 0.05). Asterisks denote significant differences between different sampled transects: * *p* < 0.05; ** *p* < 0.01; *** *p* < 0.001.

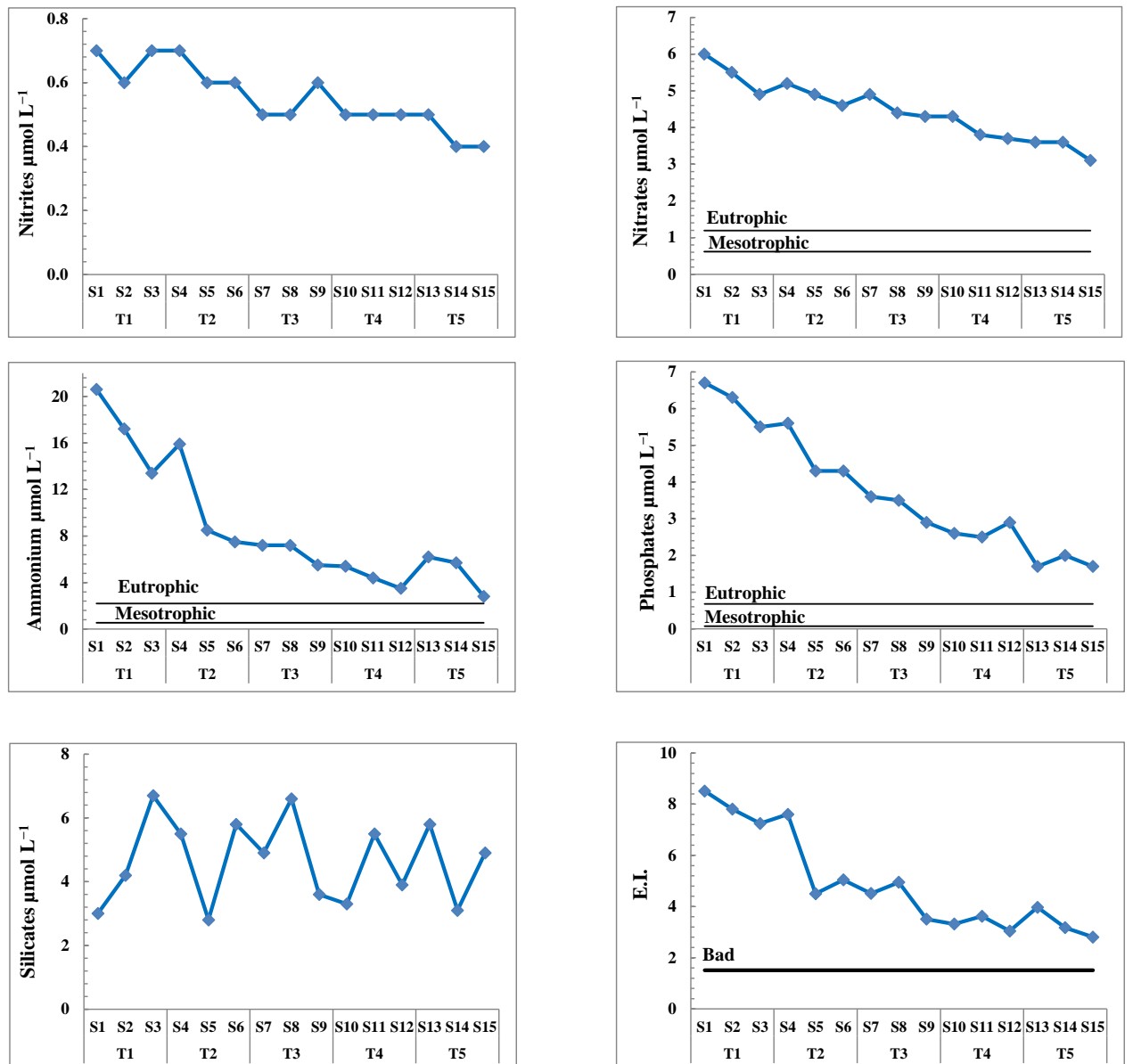

**Figure 3.** Spatial variation of nitrites, nitrates, ammonium, phosphates, silicates, and Eutrophication index (E.I.) at sampled stations (S1–S15), divided according to transects (T1–T5).

Ammonium concentrations comprised 59.60% of total dissolved inorganic nitrogen (DIN), with nitrates at 35.90% and nitrites at 4.50%. The ammonium contribution to total dissolved inorganic nitrogen (DIN) reached 75.50% at station S1.

The spatial distribution of phosphates exhibited the same surface gradient as nitrites, nitrates, and ammonium (Figure 3). Phosphate concentrations ranged from 1.7 (S12 and S15) to 6.7 (S1) µmol L$^{-1}$ (average 3.74 ± 1.65 µmol L$^{-1}$). T1 exhibited the highest amounts of phosphates, reaching 6.17 ± 0.61 µmol L$^{-1}$, which was significantly higher than those recorded in other transects ($p < 0.001$) (Table 1). In contrast, silicates did not show any gradient of variation but varied from 2.8 (S5) to 6.7 (S3) µmol L$^{-1}$ (4.64 ± 1.32 µmol L$^{-1}$) ($p > 0.05$) (Figure 3, Table 1).

All E.I. values exceeded the value 2.16, and thus they were indicative of a generalized eutrophication, with the highest value of 9.37 µmol L$^{-1}$ at S1 (Figure 3). Strong correlations (>0.8) were found between the nutrients' concentrations (ammonium, nitrates, nitrites, and phosphates). The values of E.I decreased drastically from T1 to T5 ($p < 0.01$) (Table 1).

### 3.2. Chl-a and Phytoplankton Community Structure

Chl-*a* concentration ranged from 0.04 at S15 to 0.50 µg L$^{-1}$ at S1 and S2 (average of the whole area, 0.20 ± 0.16 µg L$^{-1}$), respectively (Figure 4). It was positively correlated with ammonium ($\rho$ = 0.91; N = 15; $p$ < 0.001), phosphates ($\rho$ = 0.97; N = 15; $p$ < 0.001), and nitrates ($\rho$ = 0.98; N = 15; $p$ < 0.001). The abundance of total phytoplankton ranged from 14,600 × 10$^3$ (S11) to 98,400 × 10$^3$ (S1) cells L$^{-1}$ (Figure 4). Total phytoplankton abundances were recorded in T1 and T2, and were high and significantly similar (Table 1). In southern transect (T5), phytoplankton shows a slight increase but not significant (Table 1). Positive correlation was observed between total phytoplankton abundance and E.I. ($\rho$ = 0.60; N = 15; $p$ < 0.05).

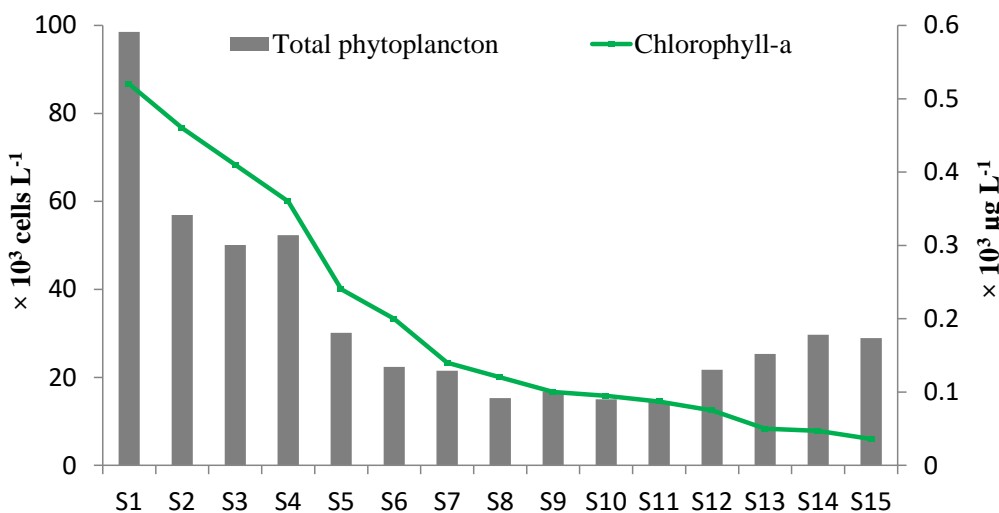

**Figure 4.** Spatial variation of abundance of total phytoplankton and Chl-*a* concentrations at 15 stations (S) grouped in 5 transects (T) in the marine coastal area of Gulf of Gabès.

A total of 42 phytoplankton *taxa* were identified, consisting of 20 Dinophyta, 18 Bacillariophyta, 2 Cyanobacteriae, 1 Euglenophyceae, and 1 Dictyochophyceae (Table 2). Four potentially toxic species of Dinophyta (*Amphidinium carterae*, *Dinophysis caudata*, *Prorocentrum lima*, *Prorocentrum minimum*) and one of Bacillariophyta (*Pseudonitzchia* sp.) were recognized.

**Table 2.** Species composition of phytoplankton collected at 15 stations in the Gulf of Gabès. *: Toxic taxa.

| Class | Order | Species |
|---|---|---|
| Bacillariophyta | Centric | *Coscinodiscus* sp. |
| | | *Skeletonema costatum* |
| | | *Thalassiosira* sp. |
| | | *Rhizosolenia styliformis* |
| | | *Leptocylindrus* sp. |
| | | *Hemiaulus* sp. |
| | | *Chaetoceros costatus* |
| | Pennates | *Grammatophora* sp. |
| | | *Licmophora* sp. |
| | | *Thalassionema nitzchoides* |
| | | *Rhabdonema* sp. |
| | | *Navicula* sp. |
| | | *Pinnularia* sp. |

**Table 2.** *Cont*.

| Class | Order | Species |
|---|---|---|
| | | *Pleurosigma simonsenii* |
| | | *Amphiprora* sp. |
| | | *Coccneis* sp. |
| | | *Nitzchia* sp. |
| | | *Pseudonitzchia* sp.* |
| Dinophyta | Gymnodiniales | *Gymnodinium* sp |
| | Dinophysiales | *Amphidinium carterae* * |
| | | *Dinophysis caudata* * |
| | Prorocentrales | *Prorocentrum compressum* |
| | | *Prorocentrum gracile* |
| | | *Prorocentrum lima* * |
| | | *Prorocentrum micans* |
| | | *Prorocentrum minimum* * |
| | | *Prorocentrum triestinum* |
| | Peridiniales | *Neocerartium candelabrum* |
| | | *Neocerartium furca* |
| | | *Neocerartium lineatum* |
| | | *Neocerartium tripos* |
| | | *Neocerartium setaceum* |
| | | *Ostreopsis ovata* |
| | | *Scrippsiella trochoidae* |
| | | *Peredinium* sp. |
| | | *Protoperedinium minatum* |
| | | *Protoperedinium depressum* |
| | | *Protoperedinium ovum* |
| Euglenophyceae | Euglenales | *Euglena acusformis* |
| Cyanobacteria | Nostocales | *Anabena flosaquae* |
| | Hormogonales | *Tichodesmium erythraeum* |
| Dictyochophyceae | Dictyochales | *Dictyocha fibula* |

Bacillariophyta were the most important group in terms of cell abundance (Figure 5), representing from 67.7% (S4) to 89.2% (S15) of total phytoplankton. The abundance of Bacillariophyta was strongly affected from one transect to another ($p < 0.01$) (Table 1). Moreover, abundance of Bacillariophyta was positively correlated with E.I. ($\rho = 0.45$; N = 15; $p < 0.05$). Cyanobacteria and Euglenophyceae were also positively correlated with E.I. ($\rho = 0.86$; N = 15; $p < 0.001$ and $\rho = 0.93$; N = 15; $p < 0.001$, respectively). Euglenophyceae was the second group representing 11.5% of total phytoplankton abundance with a maximum contribution in S1 with 20.3% (Figure 5). The abundance of this group decreased significantly from $11.87 \pm 7.08 \times 10^3$ cells L$^{-1}$ (T1) to $0.53 \pm 0.35 \times 10^3$ cells L$^{-1}$ (T5) ($p = 0.009$). Cyanobacteria *taxa* were observed only in seven stations (S1–S7) and their densities differed slightly between transects T1, T2, and T3 ($p < 0.05$) (Figure 5, Table 1). Dinophyta abundances were significantly high in T1, T2, and T5, ranging from 2.00 to $5.33 \times 10^3$ cells L$^{-1}$ (Table 1). However, they did not exceed $1.77 \times 10^3$ cells L$^{-1}$ in T3 and

T4 (Table 1). *Dictyocha fibula* was the only identified Dictyochophyceae, and it was observed only at station S12 with 100 cells L$^{-1}$.

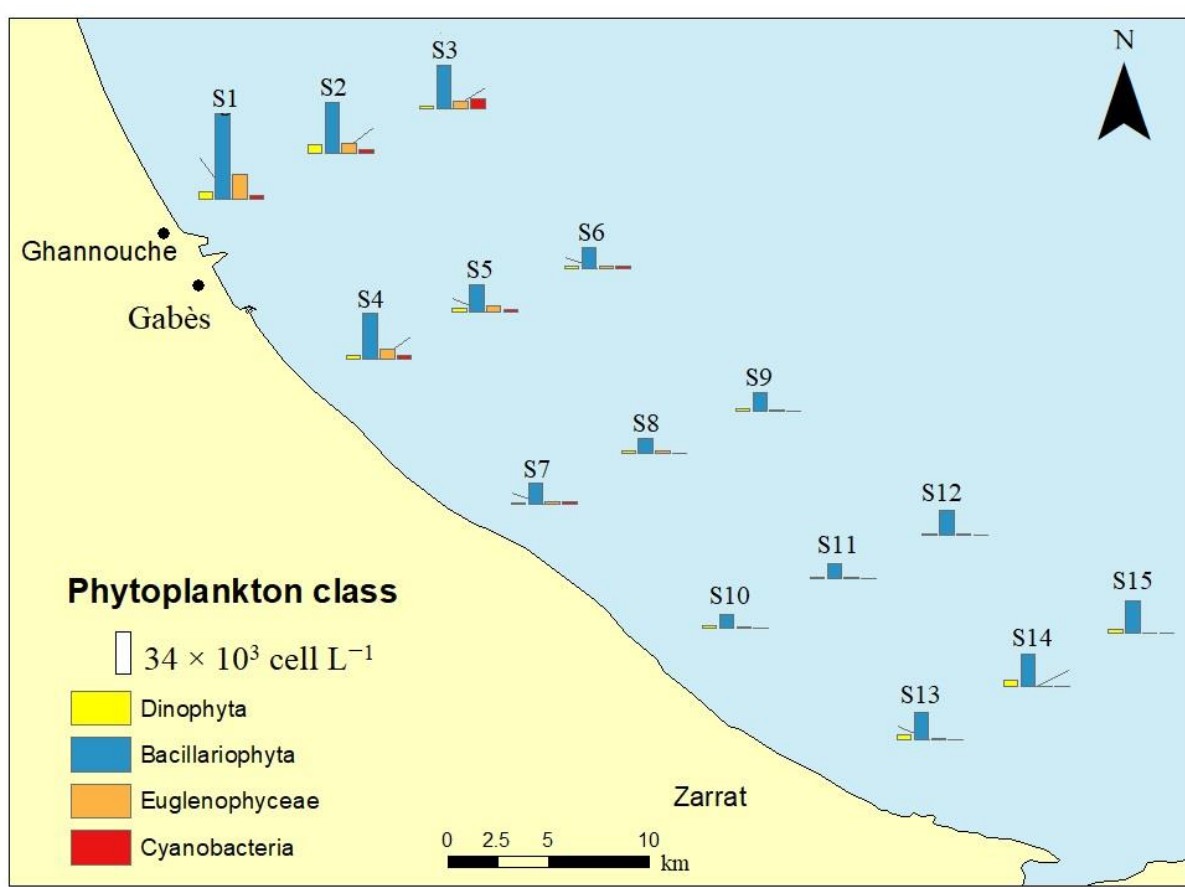

**Figure 5.** Phytoplankton class abundance at sampled stations (number beside the bar indicates the scale of abundance).

*Chaetoceros costatus*, *Euglena acusformis*, *Thalassionema nitzchoides*, and *Thalassiosira* sp. Dominated the community and seemed to be not affected by eutrophication (Figure 6). *Thalassiosira* sp., *Chaetoceros costatus*, and *Euglena acusformis* were abundant at most eutrophic stations, reaching 24,800 cells L$^{-1}$, 22,200 cell L$^{-1}$, and 20,000 cell L$^{-1}$, respectively, in S1 (Figure 6), and showed high significant positive correlations with E.I. ($\rho = 0.89$; N = 15; $p < 0.001$, $\rho = 0.94$; N = 15; $p < 0.001$ and $\rho = 0.93$; N = 15; $p < 0.001$, respectively). Figure 7 shows the pattern of dominant species, total phytoplankton, and Chlorophyll-a throughout the study. *Chaetoceros costatus*, *Euglena acusformis*, and *Thalassiosira* sp. densities roughly followed the Chl-*a* concentration. In contrast, the abundance of *Thalassionema nitzchoides* was inversely correlated to Chl-*a*. *Thalassionema nitzchoides* proliferated at eutrophic stations with less than 9000 cell L$^{-1}$ (Figure 6). This species showed a strong negative correlation with E.I. ($\rho = -0.94$; N = 15; $p < 0.001$).

The diversity index of phytoplankton (H′) ranged between 2.7 (S3) and 3.6 (S10) bits cell$^{-1}$ (Figure 6) and showed a negative correlation with E.I. ($\rho = -0.57$; N = 15; $p < 0.05$). Overall, the phytoplankton community was more diversified in transects T3, T4, and T5 and exhibited the highest values of H′ exceeding 3 bits cell$^{-1}$ (Table 1). The pattern of the Pielou index (J) followed that of H′ and ranged between 0.6 and 0.9 (Figure 6).

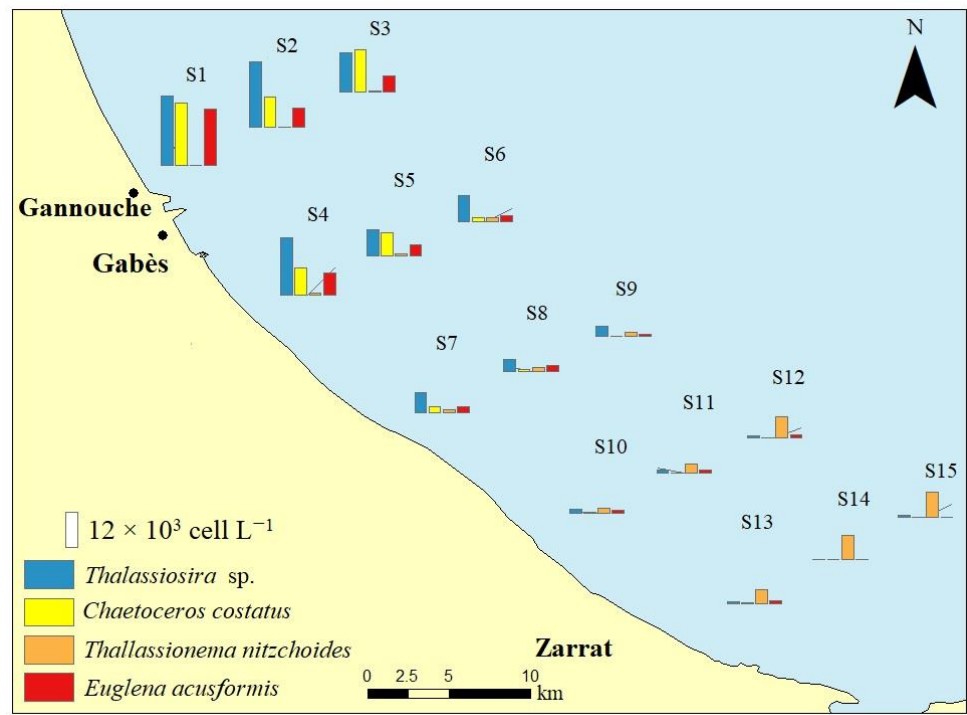

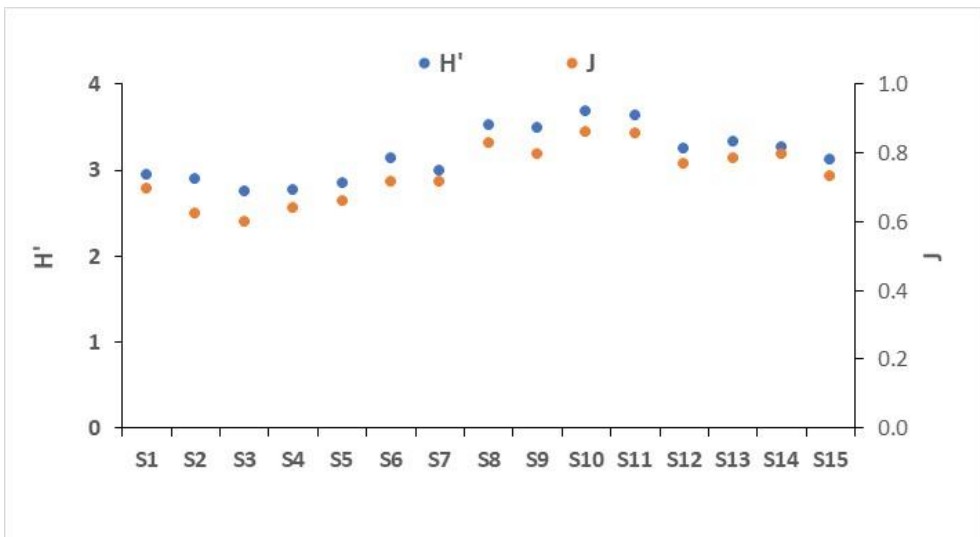

**Figure 6.** Spatial abundance of most important phytoplankton taxa (number beside the bar indicates scale of abundance) (**top**) and diversity indices (H′ and J) (**bottom**).

*3.3. Copepoda Community Structure*

A total of 24 *taxa* belonging to three orders (Calanoida, Cyclopoida, Harpacticoida) were identified (Table 3). The first two groups were the most diverse with 14 and 8 species, respectively. Harapacticoida were only represented by two species, *Euterpina acutifrons* and *Clytmnestra scutellata*. Nauplii dominated copepod communities with an average contribution between 34.9% at S8 and 74.1% at S1, (Figure 7). Their abundance ranged between 21,700 ind·m$^{-3}$ (S4) and 193,500 ind·m$^{-3}$ (S14) (Figure 7). The nauplii abundance showed a significant gradual increase from T1 to T5 (F = 36.86; $p < 0.001$) (Table 1). Spearman's test revealed that the spatial distribution of nauplii was negatively correlated with E.I. ($\rho = -0.87$; N = 15; $p < 0.001$).

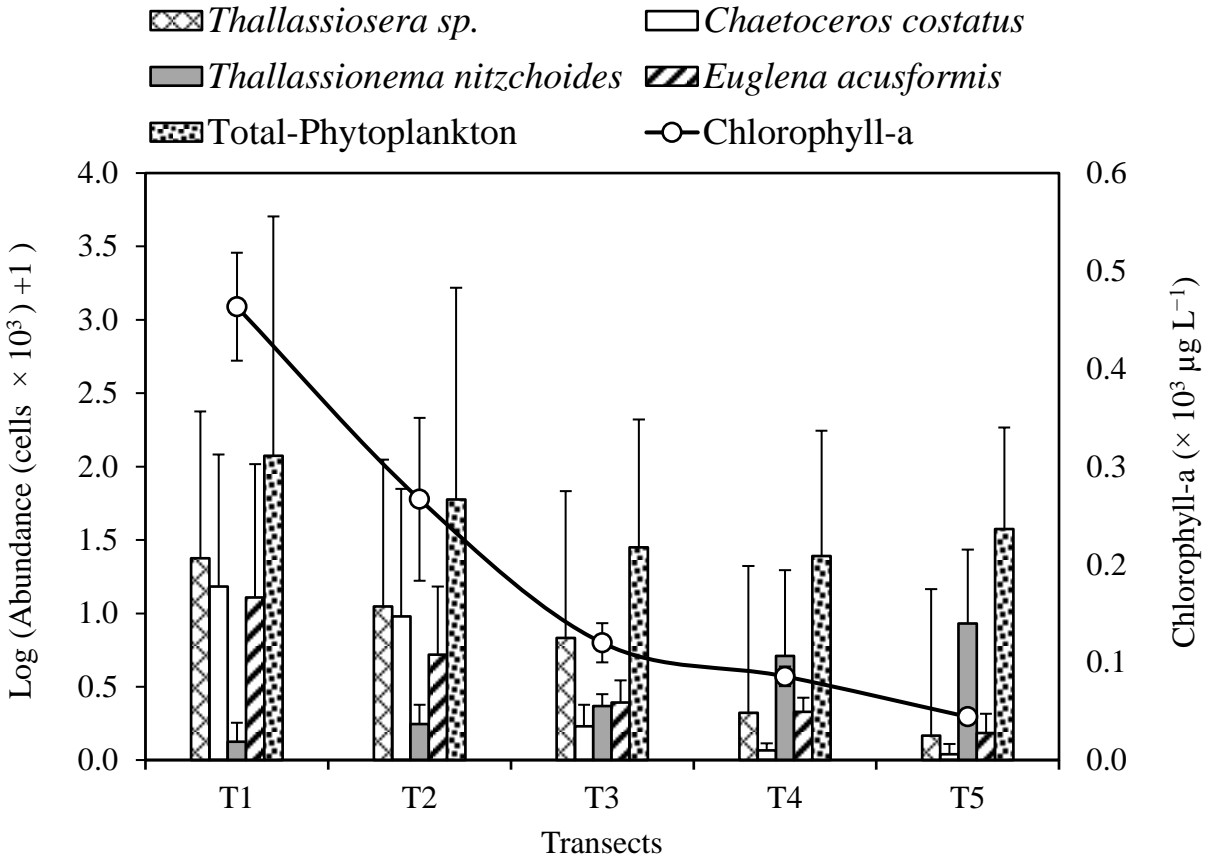

**Figure 7.** Spatial abundance of most important phytoplankton taxa, total phytoplankton, and Chlorophyll-a along the five transects.

**Table 3.** Species composition of Copepoda collected in 15 stations in the Gulf of Gabès. Taxa with an heterogeneous composition (e.g. nauplii, copepodids, adults) are not indicated.

| Order | Family | Species |
|---|---|---|
| Calanoida | Acartiidae | *Acartia clausi* |
| | | *Acartia italica* |
| | | *Acartia longiremis* |
| | | *Acartia discaudata* |
| | | *Acartia danae* |
| | | *Acartia bifilosa* |
| | | *Paracartia latisetosa* |
| | Centropagidae | *Centropages typicus* |
| | | *Centropages kroyeri* |
| | | *Centropages chierchiae* |
| | Temoridae | *Temora longicornis* |
| | | *Temora stylifera* |
| | Paracalanidae | *Paracalanus parvus* |
| | | *Paracalanus aculeatus* |

**Table 3.** *Cont.*

| Order | Family | Species |
|---|---|---|
| Cyclopoida | Oithonidae | *Oithona nana* |
| | | *Oithona similis* |
| | | *Oithona plumifera* |
| | | *Oithona helgolandica* |
| | Corycaeidae | *Corycaeus ovalis* |
| | | *Corycaeus specious* |
| | | *Corycaeus lotus* |
| | Oncaeidae | *Oncaea conifera* |
| Harpacticoida | Euterpinidae | *Euterpina acutifrons* |
| | Clytemnestridae | *Clytemnestra scutellata* |

The abundance of copepodids ranged from 3900 ind·m$^{-3}$ (S1) to 64,700 ind·m$^{-3}$ (S15) (average $\pm$ s.d. = 29,000 $\pm$ 19,200 ind·m$^{-3}$) and it varied significantly between transects ($p < 0.001$) (Figure 8, Table 1). The abundance of copepodids was strongly negatively correlated with E.I. ($\rho = -0.90$; N = 15; $p < 0.001$). The highest total adults' abundance (39,500 ind·m$^{-3}$) (average $\pm$ s.d. = 26,600 $\pm$ 11,800 ind·m$^{-3}$) was observed at S11 (Figure 8). Numbers of adults were tightly negatively correlated with E.I. ($\rho = -0.88$; N = 15; $p < 0.001$).

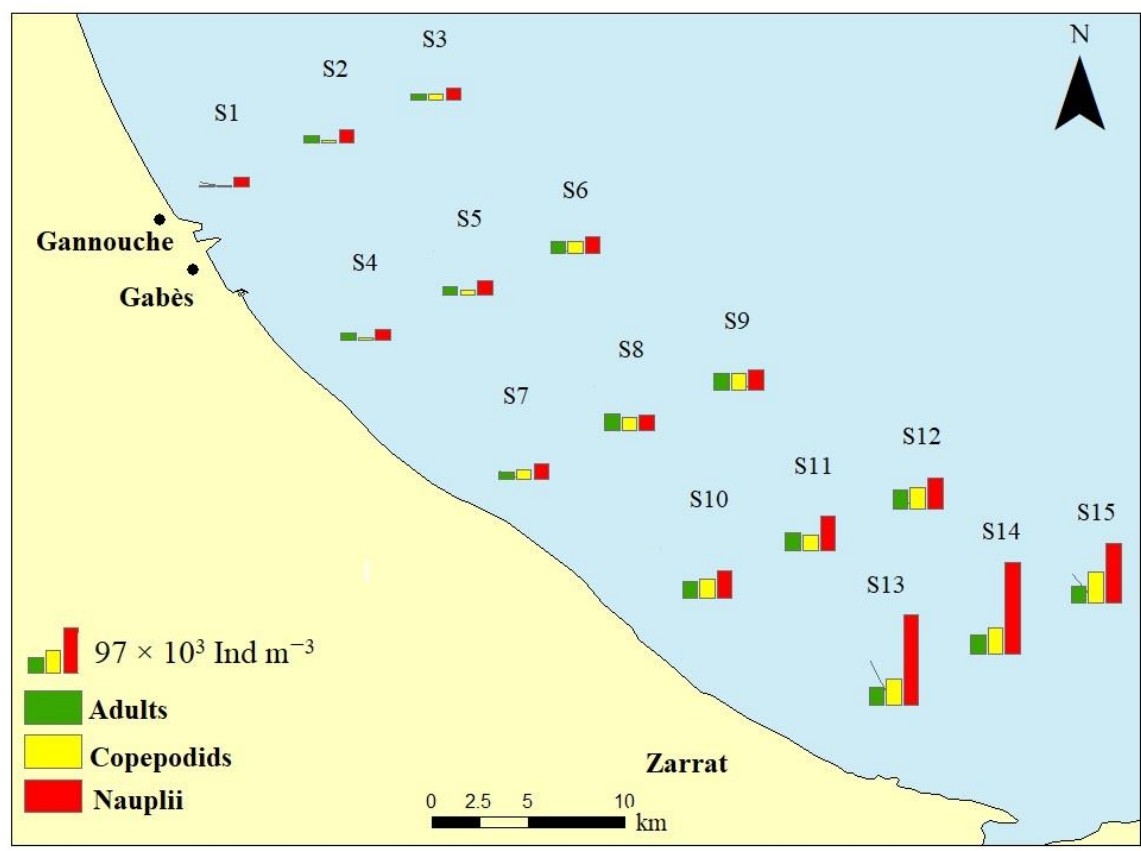

**Figure 8.** Spatial variations of copepod life-cycle stages' (nauplii, copepodids, and adults) abundance: (number beside the bar indicates scale of abundance).

Sex ratios (n. females/n. males) reported for sampling stations at Gabès coast are skewed toward females. However, at S1 and S2, sex ratio was skewed towards males (Figure 9), and such an index appeared strongly and negatively correlated with E.I. ($\rho = -0.83$; N = 15; $p < 0.001$) and positively with pH ($\rho = 0.98$; N = 15; $p < 0.001$) and with depth of the water column ($\rho = 0.77$; N = 15; $p < 0.01$).

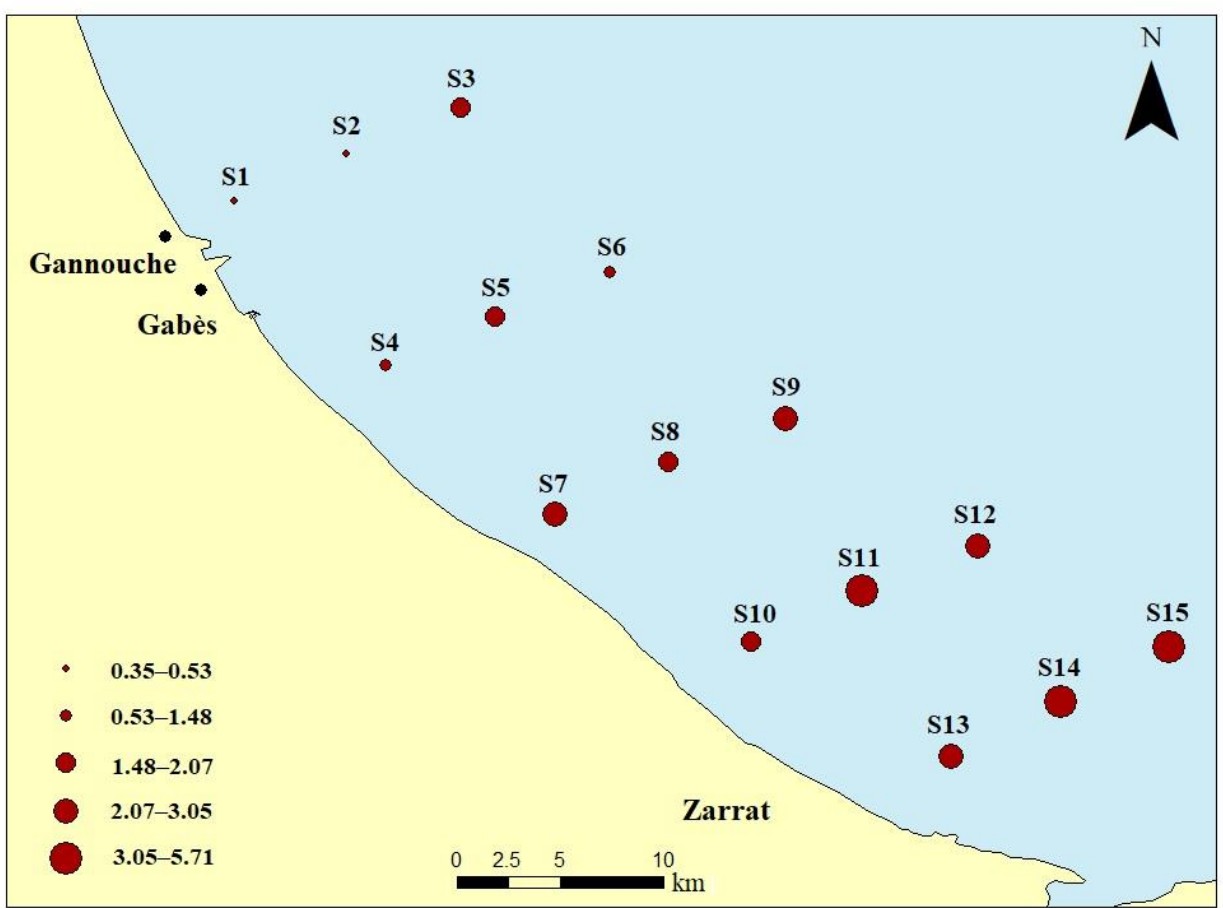

**Figure 9.** Spatial distribution of Copepoda sex ratio (Females/Males).

*Paracalanus parvus* (Calanoida) dominated the Copepoda community (Figure 10), ranging from 0 (S1 and S2) to 18,300 ind·m$^{-3}$ (S13), followed by *Oithona similis* (Cyclopoida), which reached its highest abundance at S4 with 4900 ind·m$^{-3}$ (Figure 10). *Euterpina acutifrons* (Harpacticoida) ranged from 400 ind·m$^{-3}$ (S1) to 3200 ind·m$^{-3}$ (S6 and S7), and *Acartia clausi* fluctuated between 800 ind·m$^{-3}$ at S1 and 2900 ind·m$^{-3}$ at S8 (Figure 10). Spatial distribution of *P. parvus* was negatively correlated with E.I. ($\rho = -0.84$; N = 15; $p < 0.001$) and positively with *T. nitzchoides* abundance ($\rho = 0.95$; N = 15; $p < 0.001$). The Copepoda diversity H′ ranged from 1.7 (S1) to 3.0 (S13 and S15) (ANOVA, F = 4.84, $p = 0.02$) (Figure 10, Table 1). H′ was negatively correlated with E.I. ($\rho = -0.85$; N = 15; $p < 0.001$).

### 3.4. Copepods and Phytoplankton Relationship with Environmental Conditions

The stress value for the two-dimensional MDS plot was 0.08, showing a three-group separation (Figure 11). The MDS results confirm the heterogenous distribution of both phytoplankton and Copepoda communities along the Gulf of Gabès. Cyanobacteria, Euglenophyceae, and the diatoms *C. costatus* and *Thallassiosira* sp. are positively correlated with nitrogen forms and total phosphates, constituting group C1 (stations S1, S2, S3, and S4). All copepod stages were correlated with pH, temperature, and depth forming a separate group (C2).

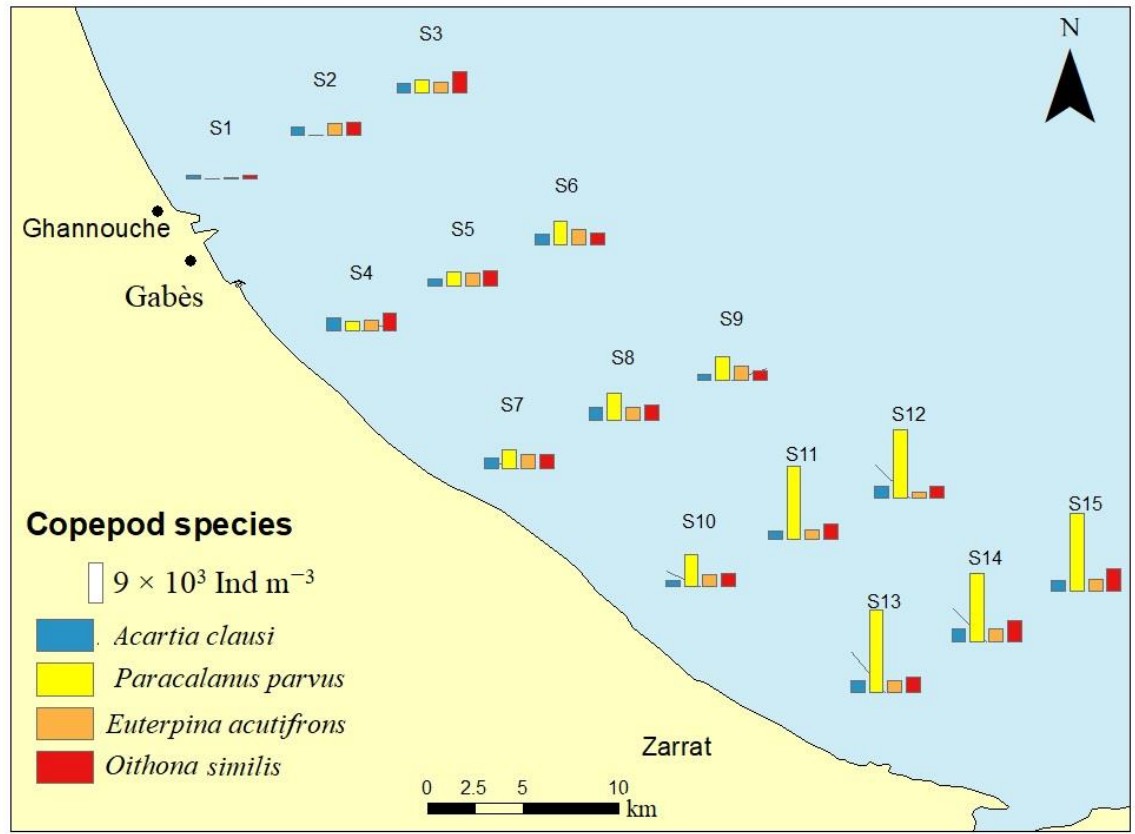

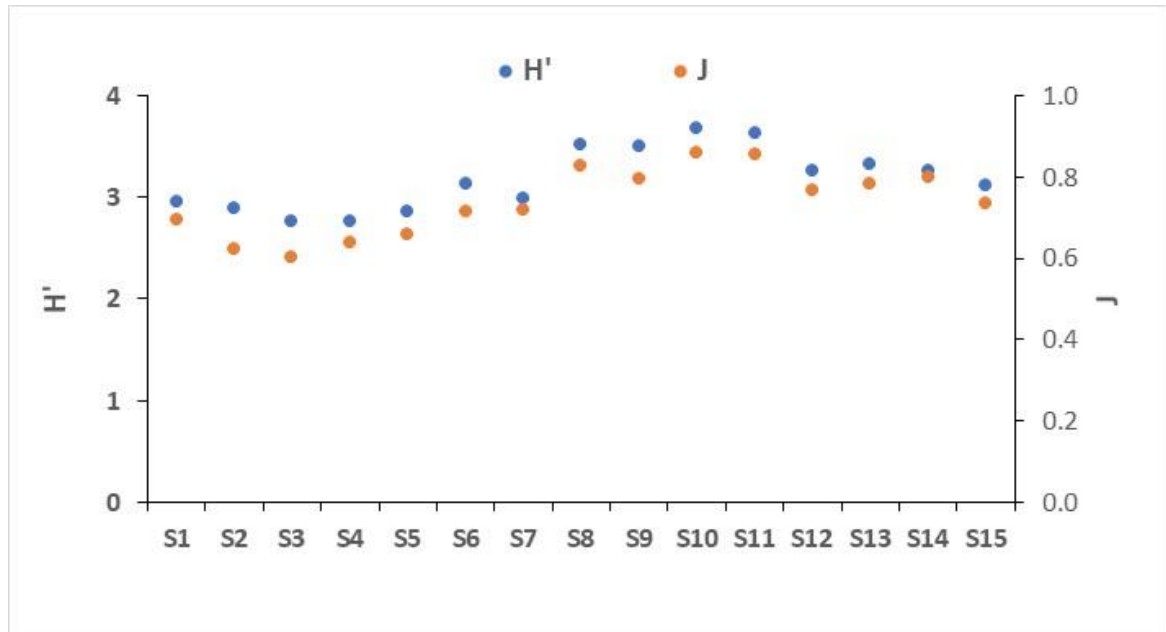

**Figure 10.** Spatial abundance of most important Copepoda taxa (number beside the bar indicates scale of abundance) (**top**) and diversity indices (H′ and J) (**bottom**).

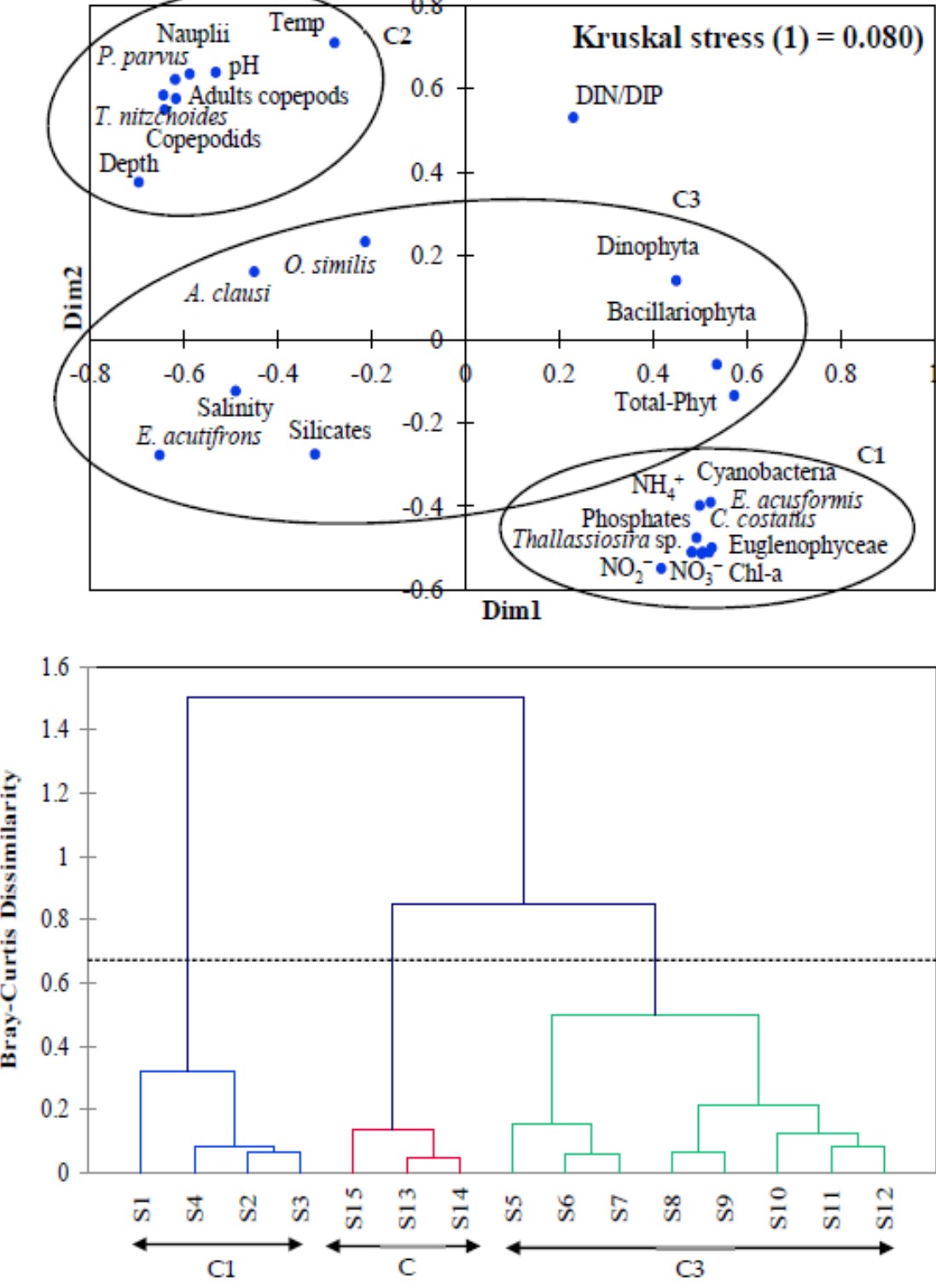

**Figure 11.** Two-dimensional MDS ordination of the groups and dominant taxa of phytoplankton (Phyt), copepods, and the physico-chemical variables. Dissimilarity among the stations was determined using the Bray–Curtis similarity index and was then superimposed on the MDS plot. The groups of stations (C1, C2, and C3) are highlighted on the dendrogram.

## 4. Discussion

This study deals with a geographic site characterized by a variable bathymetry and an arid Mediterranean climate which are someway responsible for the temperature and salinity

variability between stations. Environmental conditions in the Gulf of Gabès are affected also by anthropogenic activities. Gabès is the biggest city in the South of Tunisia (population: 131,000) and has a big industrial agrochemical complex, the Tunisian Chemical Group. In the past 40 years, several pollution studies have been regularly reported from the Gulf of Gabès, with impacts on marine organisms [23,44–49]. Anthropogenic activities are reported to increase quantities of organic and inorganic matter into the Gulf of Gabès [20,47,48,50–52], with accentuation of eutrophication.

In this study, high nutrient concentrations (nitrogen and phosphorus enrichment) were recorded at surface waters of all stations (S1–S15) along Gabès coast, thus confirming the widespread status of eutrophic area. All the E.I. values were >2.16 (average, 4.78), indicating a high ecological alteration of the marine ecosystem, according to Primpas et al. [35] and Simboura et al. [53]. The high availability of inorganic phosphate along the coast of Gabès is associable with agricultural use of land along the coast [54]. In the present study, ammonium was the main form of dissolved nitrogen. It is a typical characteristic of eutrophic waters probably derived by a non-treated anthropogenic wastewater input [55]. The Chl-*a* concentrations observed during this study (average, $0.20 \pm 0.16$ µg $L^{-1}$) are in the lower range of values previously reported for the Mediterranean Sea [56–58], and they are not in agreement with the eutrophic status of the coastal waters of Gabès. Bel Hassen et al. [59] already reported the low values of Chl-*a* in the Gulf of Gabès and interpreted it as affected by the heterotrophic character of most unicellular plankton, implicitly admitting the possibility of these organisms profiting from decomposition of organic material, in the trophic chain, more than on photosynthesis.

The human-impacted coast of Gabès City showed a comparable species richness (42 phytoplankton species in March 2014) to that on the South coast of Sfax in the same Gulf, where the phytoplankton community had consisted of 37 taxa in spring 2010 and 54 taxa in spring 2011 [60].

Abundance of Euglenophyceae and Cyanobacteria observed at stations S1–S7 correlated with the high nitrogen concentrations, as reported in other studies [61–63]. This made them strong competitors with Bacillariophyta in eutrophicated conditions, and indicator taxa of the trophic status of waters [64,65]. Phytoplankton community composition is also considered as a good bioindicator because of its rapid responses to environmental fluctuations [66]. In general, opportunistic and r strategist species (such as *Chaetoceros costatus*, *Euglena acusformis*, and *Thalassiosira* sp.) take advantage of nutrient availability which occurs with eutrophication [67–70]. In this case, the cited species could be elected as good bioindicators of eutrophication.

The Copepoda community appeared highly diverse in the Gabès area (24 taxa identified). Neritic and small planktonic copepods of the coastal waters of Gabès were characterized by species considered tolerant: *A. clausi*, *E. acutifrons*, and *O. similis*, but, in any case, suffering eutrophication (because their abundance is inversely correlated with E.I.). The adoption of a successful reproductive strategy combined with an omnivorous diet, lower metabolic needs, and tolerance to pollution are certainly behind the prominence of small planktonic copepods in the Gulf of Gabès [71]. *P. parvus*, the most abundant copepod, was absent from the most eutrophic stations (S1 and S2). Furthermore, Gubanova et al. [72] found a drastic drop in *P. parvus* abundance following two decades of pollution and eutrophication in Sevastopol Bay (Black Sea). Nauplii and copepodids showed a general avoidance of the most eutrophic stations and they negatively correlated with E.I.

The results of sex ratio, among adult Copepoda, suggest that males could be more tolerant to eutrophication and pollution than females because they predominate at the most eutrophic stations S1 and S2. For marine Copepoda, sex ratio is governed by diverse parameters, such as pH, chemical composition, temperature, depth, and hydrostatic pressure, which leads to an alteration of sex ratio and reproductive timing [73,74]. Our results confirm the impact of eutrophication and pH on sex ratio. It is also noteworthy that S1 and S2 are characterized by high levels of trace metals [27].

The diversity indices (H′ and J) of phytoplankton and Copepoda showed negative correlations with E.I. Therefore, the relatively high H′ values for phytoplankton suggest a composition of the community based on species well-adapted to profit from eutrophic water in the area of Gabès.

## 5. Conclusions

This study used the high eutrophication levels of the Gulf of Gabès, arranging accordingly a spatial gradient with growing values of E.I. from Zarrat to Gannouche, to establish if a correlation exists with plankton composition. High nutrient loading from expanding human presence and activities in the coast of Gabès must be carefully considered in order to propose a recovering of the natural trophic status. An integrated management of the coastal marine environment of Gabès City remains a necessity. The present study offers the possibility to use plankton composition in defining the efficacy of recovery actions. Eutrophication seems to affect deeply the abundance of the zooplankton, which show absence of some species (sensible) and abundance diminution in highly eutrophic stations also for the species considered "tolerant". In the heavily eutrophic stations, even the sex ratio of adults was reverted. Phytoplankton, in general, showed a higher tolerance than zooplankton. Among the phytoplankton, tolerant and adapted species (*Chaetoceros costatus*, *Euglena acusformis*, and *Thalassiosira* sp.) sustained both abundance and diversity scores in the most eutrophic stations, also favored by the absence of the graze pressure of zooplankton. Among zooplankton, on the contrary, the numerical abundance was inversely correlated with the eutrophic gradient, and even those species considered as "tolerant" were absent from the more eutrophic stations.

**Author Contributions:** Conceptualization, N.A.-T. and W.G.; methodology, N.A.-T.; software, Y.A.-E. and W.G.; validation, G.B. and V.L.; writing—original draft preparation, N.A.-T.; writing—review and editing, V.L., W.G., M.A. and Q.K.; visualization, H.A.; supervision, G.B. All authors have read and agreed to the published version of the manuscript.

**Funding:** This research received no external funding.

**Acknowledgments:** We would like to thank Mariem Fessi and Amal Baccar for technical help.

**Conflicts of Interest:** The authors declare no conflict of interest.

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
