# Peer review of "Effects of Eutrophication on Plankton Abundance and Composition in the Gulf of Gabès (Mediterranean Sea, Tunisia)"

_water, doi:10.3390/w14142230_

Round 1
Reviewer 1 Report
This study investigated the environmental and ecological effects of the intensive human activities (e.g., agrochemical and wastewater pollutions) on the Gulf of Gabès (Mediterranean Sea, Tunisia). This study is in accordance with previous studies that confirm the high eutrophication levels of the sea of Gabès. Moreover, this study suggests that the opportunistic and r strategist species (Chaetoceros costatus, Euglena acusformis, and Thalassiosira sp.), the Shannon-Weaver diversity of phytoplankton and Copepoda, as well as Copepoda sex-ratio can be used as the parameters to assess the status of eutrophication of the Gulf of Gabès, which might be useful in the management of the coastal environments. Generally speaking, this manuscript is well written.
More comments about the manuscript are listed below.
Line 81-87 These two paragraphs can be combined into one paragraph, and at the end of this introduction, it might be better, if some hypotheses have been proposed.
Line 227-235 Combine these three paragraphs into one paragraph.
Author Response
We thank the reviewer for the extremely positive judgement of the manuscript.
We also corrected some parts accordingly with his suggestion

Reviewer 2 Report
This manuscript aims to examine the effects of eutrophication on phytoplankton and copepod abundance and composition in Gulf of Gabès, but I found this manuscript was not well-structured and the analysis methods were also inappropriately. Furthermore, the main findings of this manuscript were regular and lacked of scientific novelty. My major comments are as follows:
- The manuscript is based on a small sampling effort (only 15 sampling sites in a single investigation). Is such low level of sampling effort sufficient to represent the assemblage structure of phytoplankton and copepod in the studied gulf? I strongly doubt this point.
- The results (e.g., the Fig. 2-8 only simply displayed the data) is too descriptive and lacks of necessary and appropriate statistical analysis. For example, it seems the concentrations of nutrients differ significantly across the 15 sites. Can they be classified into different site groups subjected to different levels of eutrophication? If such classification is available, the author can used appropriate methods (e.g., ANOVA for environmental variables, density and diversity indices, NMDS and ANOSIM for assemblage composition) to analyze their data.
- I strongly recommended the authors used figures or table to show the correlation between diversity indices and eutrophication index. The authors also need to use more appropriate analysis to examine the diversity indices-environment (e.g., multiple regression analysis) and assemblage-environment (e.g., constrained ordination analysis, RDA or CCA) relationships.
- Some results and analysis are unnecessary or do not make sense. For example, the results of fig. 2-8 should be reanalyzed and reorganized. There is no sense to assess the correlation between Bacillariophyta abundance and total phytoplankton abundance (line 189-191), as Bacillariophyta were the most dominate group of phytoplankton.
- The text in the manuscript was well structured and some paragraphs were hard to follow. For example, the abstract lacks of ecological significance but with too many unnecessary sentences. For the Introduction section, the eutrophication situation of study area should be incorporated into the same paragraph, while the response of plankton to eutrophication should be placed in the other paragraph. The fourth paragraph was difficult to connect to the context (line 81-83).
Author Response
we answer the question risen by the reviewer hoping to have been satisfactorily.
Yes, we admit that a paper on an eutrophy mapping and a zooplankton analysis probably announces an ecological content more important than that presented. but we carefully wanted to avoid such a classification of our study in the frame of field ecology researches, and clarified already in the title the true aim of the research: to study a correspondence between two variables (the E.I. and the plankton), and reducing to the maximum the interference of other enrvironment variables (water column, season, dissolved oxygen, and other environment descriptors).
All the suggestions have been considered and answered, in the frame of this our point of view.

Reviewer 3 Report
The manuscript submitted by Annabi-Trabelsi and collaborators addresses a very interesting ecological problem in the Mediterranean Sea. In fact, the authors analyze the effects of eutrophication on plankton abundance and composition (phytoplankton and copepod zooplankton) in an industrialized area of Tunisia. The study is weak because only one sampling date has been selected (spring 2014); being well known that studies on eutrophication must be based on at least one annual cycle. However, the information is very interesting and with a high importance for future restoration actions in Gabès coast, so and I believe that after major changes it could be accepted for publication in Water.
In this sense, the authors should explain:
(i) As in any scientific work a hypothesis must be formulated
(ii) Why samples for physico-chemical analyses, Chl-a and for analysis of phytoplankton were collected only at 1 m below the sea level?
(iii) Why they do not analyze TP and TN, the typical nutrient forms used in eutrophication indexes (such as TSI – Carlson 1977)? Why they do not use other eutrophication indexes (TSI), for example employing Secchi disk or Chla-a to evaluate eutrophication? Other interesting index is for example, the composite trophic status index (TRIX) that provides useful metrics for the assessment of the trophic status of coastal waters. It was originally developed for Italian coastal waters and then applied in many European seas (Adriatic, Tyrrhenian seas).
(iv) The statistical analysis could be improved and thus obtain more information. For example, I propose the use of GLM or CCA, which would allow us to clearly see the effect of environmental variables on biotic communities. In any case, Pearson's correlation analysis requires normal data and no tests have been performed on this. Due to the nature of the data, I believe that these are not normal, so the correlation analyses should be carried out with the Spearman correlation index.
(v) In addition to the Shannon index, it would be very interesting to show the Pielou index, which gives very interesting information about the equitability of the community.
(vi) When talking about the Shannon index, it would be necessary for future comparisons in others manuscripts to indicate how it has been calculated (log10 - ln or log2)
(vii) It would be of great interest for future work to indicate the species that appear in each season. This information should appear in Table 1 for phytoplankton and Table 2 for copepods.
(viii) Although the sex ratio can fluctuate considerably in a population throughout the year (Devreker et al., 2010), and the same spatially (Zadereev & Tolomeyev, 2007), I believe that more information can be obtained if they are compared (Spearman rank correlations) the values of this sex ratio with the environmental variables.
(ix) Discussion should be greatly improved.
The authors indicate in the introduction that the study area is an area with high industrialization. Any comment in this sense, about possible contamination of the waters and its effect on the planktonic community should be commented on; especially since perhaps not all the effects detected are a consequence of eutrophication.
Author Response
we thank the reviewer for the rich discussion and the given suggestion.
as we indicated in the annexed file, our intent was the finding of a correlation between E.I. and Plankton, and this "authorized" us to avoid any other additional data. This point of view (not a study on the coastal environment; not a study of the plankton community) has been considered in restricting our focus on plankton in 15 different conditions (stations) of eutrophication.
In any cases all the suggestions have bee considered and answers were produced for each point

Round 2
Reviewer 2 Report
The revised ms was somewhat improved compared to the first version, but there are still several shortcomings and unsolved comments
1. The current results section are still too descriptive, full of descriptive figures but lack of necessary and appropriate analysis:
(1) the current fig. 2 and fig. 3 should be combined in to a table showing the differences in water quality variable among the five transects. Although the the concentrations of N and P salts are high across the 15 sites, I found the nutrient concentrations are quite different among the five transects (e.g., T1 was most eutrophic whereas T5 is least). I still recommend the authors used ANOVA to compare the differences in environmental variables and biological metrics among the five transects.
(2) the fig.4 and fig.6 also should be replaced by a table or figure (e.g., Bar-errors plots) showing the differences in biological metrics among the five transects.
(3) the display of fig.5, Fig.6 top, fig. 7 and fig.9 top are interesting, but I think they are still a bit too descriptive. Why not the authors used NMDS or other analysis to show the differences in community composition more comprehensively
(4) I cannot see the relevance of fig. 8 (Copepoda sex-ratio) to your research topic. Does previous publications reported that eutrophication would affect the sex-ratio of plankton? This point should be present in the Introduction section if the authors would like to examine this.
(5) fig. 9 bottom also should be revised as above comments.
2. the statistical analysis section is too simple and lack of many necessary information. For example, if you applied a CCA or RDA, a preliminary detrended correspondence analysis (DCA) is needed (gradient length >4 standard units indicated a unimodal model, i.e., CCA would best fit the data) Prior to analysis, whether you transform (e.g., log-transformed) your biological data and check the normality and correlations of your environmental data? Whether you use Forward selection to select the key explanatory variables?
Based on the current Fig. 10, I found the authors combined the species density data and diversity indices into a biological data. Such approach is certainly inappropriate and the authors cannot run CCA like this. The authors should only use the density data as biological data and use multiple regression and other analysis to examine the relationships between diversity indices and environmental variables
It is also strange the authors only used three environmental variables? Why not use all water quality variables
There are also several errors in the section 3.4, e.g., not the PCA axis but the first and second axis of CCA. I recommend the authors rerun the CCA (or RDA) and look other publications involving the CCA to display the results of CCA.
Author Response
The revised ms was somewhat improved compared to the first version, but there are still several shortcomings and unsolved comments
- The current results section are still too descriptive, full of descriptive figures but lack of necessary and appropriate analysis:
(1) the current fig. 2 and fig. 3 should be combined in to a table showing the differences in water quality variable among the five transects. Although the the concentrations of N and P salts are high across the 15 sites, I found the nutrient concentrations are quite different among the five transects (e.g., T1 was most eutrophic whereas T5 is least). I still recommend the authors used ANOVA to compare the differences in environmental variables and biological metrics among the five transects.
We agree with this point and have added a table (Table 1) showing the differences in water quality variables among the five transects by using the one way anova completed by the post-hoc test tukey.
(2) the fig.4 and fig.6 also should be replaced by a table or figure (e.g., Bar-errors plots) showing the differences in biological metrics among the five transects.
We have decided to keep Fig 4 and Fig 6 because they show the differences between sampled stations, and we have added histogram with bar errors illustrated the differences of total phytoplankton, the dominant groups of phytoplankton and Chl-a among five transects.
(3) the display of fig.5, Fig.6 top, fig. 7 and fig.9 top are interesting, but I think they are still a bit too descriptive. Why not the authors used NMDS or other analysis to show the differences in community composition more comprehensively
We agree that better use the correspondence Analysis that would be more descriptive and precise and have taken your advice and according to your comments on statistical analyses below we have replaced the CCA by MDS analysis completed by dendrogram based on Bray-Curtis similarity.
(4) I cannot see the relevance of fig. 8 (Copepoda sex-ratio) to your research topic. Does previous publications reported that eutrophication would affect the sex-ratio of plankton? This point should be present in the Introduction section if the authors would like to examine this.
We add: Copepod survival, fitness, sex-ratio and condition are negatively correlated with Cyanobacteria (Engström-Öst, 2015) that characterized eutrophic marine ecosystem (O'Neil, 2012). Copepod male survival may be more successful than female in eutrophic conditions and positive correlation between phosphates and the sex ratio in copepod was observed by Krupa (2005)
(5) fig. 9 bottom also should be revised as above comments.
ok
- the statistical analysis section is too simple and lack of many necessary information. For example, if you applied a CCA or RDA, a preliminary detrended correspondence analysis (DCA) is needed (gradient length >4 standard units indicated a unimodal model, i.e., CCA would best fit the data) Prior to analysis, whether you transform (e.g., log-transformed) your biological data and check the normality and correlations of your environmental data? Whether you use Forward selection to select the key explanatory variables?
Based on the current Fig. 10, I found the authors combined the species density data and diversity indices into a biological data. Such approach is certainly inappropriate and the authors cannot run CCA like this. The authors should only use the density data as biological data and use multiple regression and other analysis to examine the relationships between diversity indices and environmental variables
We agree with you and we made the MDS analysis without diversity indices
It is also strange the authors only used three environmental variables? Why not use all water quality variables
We have integrated in the MDS all aquatic variables
There are also several errors in the section 3.4, e.g., not the PCA axis but the first and second axis of CCA. I recommend the authors rerun the CCA (or RDA) and look other publications involving the CCA to display the results of CCA.
We have replaced CCA analysis by MDS as described above
Reviewer 3 Report
I am grateful that the authors have addressed or justified the majority of the indicated questions. I think the manuscript has improved and can now be published in Water. However, I have detected some small errors that must be corrected before its definitive acceptance:
(1) line 273 Change PC1 and PC2 to CCA1 and CCA2
(2) line 274: Change "The 1st Principal Component axis" to "The 1st Canonical Correspondence axis"
(3) line 286: remove trailing double parentheses
(4) line 295: I think temperature and salinity should be lowercase
(5) Results: Perhaps it would be interesting to indicate that in the CCA, the community structure indices are spatially associated with the stations with less eutrophication (lower right quadrant of the CCA).
(6) Discussion: This has been the least improved section. I think the authors should make more effort. For this reason I suggest that, in order to give a certain applicability to the work carried out and taking into account the results obtained; It would be interesting to add a paragraph to the discussion indicating that the restoration actions in the Gulf of Gabès should begin with stations S1-S4, which are the ones that the CCA analysis has indicated as the most eutrophic.
Author Response
Comments and Suggestions for Authors
I am grateful that the authors have addressed or justified the majority of the indicated questions. I think the manuscript has improved and can now be published in Water. However, I have detected some small errors that must be corrected before its definitive acceptance:
(1) line 273 Change PC1 and PC2 to CCA1 and CCA2
We change CCA by MDS
(2) line 274: Change "The 1st Principal Component axis" to "The 1st Canonical Correspondence axis"
We change by MDS
(3) line 286: remove trailing double parentheses
Done
(4) line 295: I think temperature and salinity should be lowercase
Done
(5) Results: Perhaps it would be interesting to indicate that in the CCA, the community structure indices are spatially associated with the stations with less eutrophication (lower right quadrant of the CCA).
We change CCA by MDS
(6) Discussion: This has been the least improved section. I think the authors should make more effort. For this reason I suggest that, in order to give a certain applicability to the work carried out and taking into account the results obtained; It would be interesting to add a paragraph to the discussion indicating that the restoration actions in the Gulf of Gabès should begin with stations S1-S4, which are the ones that the CCA analysis has indicated as the most eutrophic.
We cannot base restoration suggestion on the present data (only one date), as some criticism told us (correctly). This notwithstanding, the results of correlations among E.I. and plankton composition are encouraging for future and deeper studies which will adopt plankton composition as indicator of eutrophic conditions.